



**Improving Madden–Julian Oscillation Simulation in Atmospheric General**
**Circulation Models by Coupling with Snow–Ice–Thermocline One-dimensional**
**Ocean Model**
Wan-Ling Tseng[1], Huang-Hsiung Hsu[1*], Yung-Yao Lan[1], Chia-Ying Tu[1], Pei-Hsuan
Kuo[2], Ben-Jei Tsuang[3], Hsin-Chien Liang[1]
[1]Research Center for Environmental Changes, Academia Sinica, Taipei, Taiwan
[2]Center Weather Bureau, Taipei, Taiwan.
[3]National Chung-Hsing University, Taichung, Taiwan.
Corresponding author: Huang-Hsiung Hsu (hhhsu@gate.sinica.edu.tw)





**Abstract**
A one-column turbulent kinetic energy–type ocean mixed-layer model Snow–Ice–
Thermocline (SIT) when coupled with three atmospheric general circulation models
(AGCMs) to yielded superior Madden–Julian Oscillation (MJO) simulation. SIT is
designed to have fine layers similar to those observed near the ocean surface and therefore
can realistically simulate the diurnal warm layer and cool skin. This refined discretization
of the near ocean surface in SIT provides accurate sea surface temperature (SST)
simulation, thus facilitating realistic air–sea interaction. Coupling SIT with European
Centre Hamburg Model, Version 5 (ECHAM5); Community Atmosphere Model, Version
5 (CAM5); and High Resolution Atmospheric Model (HiRAM) significantly improved
MJO simulation in three coupled AGCMs compared with the AGCM driven with
prescribed SST. This study suggests two major improvements to the coupling process.
First, during the preconditioning phase of MJO over Maritime Continent (MC), the over
underestimated surface latent heat bias in AGCMs can be corrected. Second, during the
phase of strongest convection over MC, the change of the intraseasonal circulation in the
meridional circulation is the dominant factor in the coupled simulations relative to the
uncoupled experiments. The study results indicate that a fine vertical resolution near the
surface, which better captures temperature variations in the upper few meters of the ocean,
considerably improves different models with different configurations and physical
parameterization schemes; this could be an essential factor for accurate MJO simulation.
**Keywords:** Madden–Julian Oscillation, coupling, warm layer





**Short summary (plain text)**
We show that coupling a high-resolution one-column ocean model to three
atmospheric general circulation models dramatically improves Madden–Julian oscillation
(MJO) simulations. It suggests two major improvements to the coupling process in
preconditioning phase and strongest convection phase over Maritime Continent. Our
results demonstrate a simple but effective way to significantly improve MJO simulation
and potentially also seasonal to subseasonal prediction.





## 1 Introduction

The Madden–Julian Oscillation (MJO) is the dominant pattern of atmospheric intraseasonal variability in the tropics (Madden and Julian 1972; Zhang 2005; Jiang et al. 2020). It has been reported that the MJO convection is most often observed over sea surface temperature (SST) of >28°C in the Indo-Pacific warm pool (Salby and Hendon 1994). The MJO is an eastward-propagating ocean–atmosphere and convection–circulation coupled phenomenon that lasts for 20–100 days. On these timescales, low-level moisture convergence, warm SST, and shallow upper-ocean mixed-layer depth precede the eastward propagation of organized deep convection by approximately 10 days; opposite conditions follow by approximately 10 days (Krishnamurti et al. 1988; Hendon and Salby 1994; Woolnough et al. 2000). Heat flux exchange between the atmosphere and ocean modulates the intraseasonal oscillation (Shinoda and Hendon 1998; Shinoda et al. 1998). Studies have emphasized the importance of moisture and heat flux feedback in MJO (Sobel et al. 2008, 2010; DeMott et al. 2015). Besides, the MJO and oceanic wave dynamics are also suggested such as zonal wind stress anomalies associated with the MJO force eastward-propagating oceanic equatorial Kelvin wave (Hendon et al. 1998; Webber et al. 2010), and the signals could extend as deep as 1500 m in the ocean (Matthews et al. 2007). Furthermore, the westward-propagating oceanic equatorial Rossby wave can initiate the next MJO in the Indian Ocean (Webber et al. 2010; Webber et al. 2012). Evidence of oceanic intraseasonal signals coupling with atmospheric signals was observed in terms of the sea level, surface heat flux, salinity, and temperature during field experiments and in situ monitoring (Oliver and Thompson 2011; Drushka et al. 2012; Wang et al. 2013; Chi et al. 2014; DeMott et al. 2015; Fu et al. 2015).





Recent modeling studies evaluating the mechanism of ocean–atmosphere coupling
have indicated that most coupled models could improve MJO simulations but that the
ocean driven by the atmosphere contributes indirectly through improvement in the mean
state, heat flux, fresh water, and momentum. DeMott et al. (2016) estimated that direct
SST-driven ocean feedback contributes MJO propagation up to 10% by change in column
moisture. A comparison of the direct and indirect effects of SST indicated that direct
effects such as SST-driven surface fluxes tend to offset wind-driven fluxes (DeMott et al.
2015; DeMott et al. 2016; DeMott et al. 2019). The key factor of indirect ocean feedback
in the atmospheric physical process, such as strong MJO convection can amplify the
radiative feedback to MJO convections associated with large cloud system (Del Genio
and Chen 2015), the SST gradients can dive the MJO low-level convergence (Hsu and Li
2012; Li and Carbone 2012), or destabilize lower tropospheric enhance low-level
convergence to east of MJO convergence (Wang and Xie 1998; Marshall et al. 2008;
Benedict and Randall 2011; Fu et al. 2015). Many observational and model studies have
reported that coupled feedback enhances the MJO with strong horizontal moisture
advection, driven by sharp mean near-equatorial meridional moisture gradients (DeMott
et al. 2015; Jiang et al. 2018; DeMott et al. 2019; Jiang et al. 2020). These finding suggest
that high-frequency SST perturbations could improve moisture convergence efficiency
and enhance MJO propagation through relatively smooth background moisture
distribution.
Tseng et al. (2015) identified the key role of the upper-ocean warm layer in
improving the MJO eastward propagation simulation by using the European Centre
Hamburg Model, Version 5 (ECHAM5), coupled with the one-column ocean model
named Snow–Ice–Thermocline (SIT). Many observational (Drushka et al. 2012; Chi et

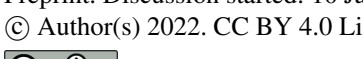



al. 2014) and modeling (Klingaman and Woolnough 2013; DeMott et al. 2019;
Klingaman and Demott 2020) studies have supported this hypothesis. However, coupling
the SIT to only one atmospheric general circulation model (AGCM) may be insufficient
to prove the effect of the coupling. In the current study, we coupled the SIT to three
AGCMs: European Centre Hamburg Model, Version 5 (ECHAM5); Community
Atmosphere Model, Version 5 (CAM5); and High Resolution Atmospheric Model
(HiRAM). As well as one additional high-resolution forecast model from Central Weather
Bureau, Taiwan (CWBGFS) to demonstrate that the improvement of MJO simulation
through coupling the upper-ocean warm layer is AGCM independent. Furthermore, we
discussed the coupling mechanism that leads to simulation improvement. Models, the
experimental design, and observational data are described in Section 2. Section 3 presents
the results, followed by a discussion in Section 4.
**2 Models, experiments, and observational data**

Observational data used in this study include precipitation from Global Precipitation

Climatology Project V1.3 (GPCP, 1° resolution) (Adler et al. (2003), outgoing longwave
radiation (OLR, 1° resolution) (Liebmann (1996)), and daily SST (Optimum Interpolated
SST, 0.25° resolution) (Banzon et al. (2014)) from the National Oceanic and Atmosphere
Administration, and variables were obtained from the European Centre for Medium-range
Weather Forecast Reanalysis-interim (Dee et al. 2011). We used a 22-year ERA-Interim
from 1989 to 2010 and a 14-year GPCP dataset from 1997 to 2010. Oceanic observational
data include those from the NCEP Global Ocean Data Assimilation System (GODAS)
(Behringer and Xue (2004) provided by the NOAA/OAR/ESRL PSL, Boulder, Colorado,





USA (https://psl.noaa.gov/data/gridded/data.godas.html) and in situ temperature profiles
from the Tropical Ocean Global Atmosphere program (McPhaden et al. 2010).
In this study, we coupled the SIT one-column ocean model (Tu and Tsuang 2005;
Tsuang et al. 2009) to four AGCMs. SIT simulates variations in the SST and upper-ocean
temperature, including the diurnally varying cool skin and warm layer in the upper few
meters of the ocean and the turbulent kinetic energy (TKE) (Gaspar et al. (1990)) in the
water column (Tu and Tsuang 2005; Tsuang et al. 2009; Tseng et al. 2015). The four
AGCMs used here are as follows. (1) ECHAM5, a the fifth-generation AGCM developed
at the Max Planck Institute for Meteorology (Roeckner 2003; Roeckner et al. 2006). It is
a spectral model employing the Nordeng (Nordeng 1994) cumulus convective scheme.
We used a horizontal resolution of T63 (approximately 2°) with 31 vertical layers and a
model top at 10 hPa (approximately 30 km). (2) NCAR CAM5 in Community Earth
System Model, version 1.2.2 (Hurrell et al. 2013) from the National Center for
Atmospheric Research. (3) HiRAM, developed based on Geophysical Fluid Dynamical
Laboratory global atmosphere and land model AM2 (Team et al. 2004; Zhao et al. 2009)
with few modifications (Chen et al. 2019). We also used CWBGFS, the second-
generation global forecast system at the Central Weather Bureau in Taiwan (Liou et al.
1997), which employs the cumulus convective scheme of Nordeng (1994), shallow
convective scheme of (Li and Wang 2000), and boundary layer of Hong and Pan (1996).
In this study, we applied 42 vertical layers in SIT, with 12 layers in the upper 10 m.
In the upper 10 m, fine resolution was designed to realistically simulate the upper-ocean
warm layer, including a layer at 0.05 mm, reproducing the cool skin of the ocean surface.
Notably, coupling of a high-vertical-resolution TKE ocean model with an AGCM is
unconventional. To account for neglected horizontal processes, the model ocean was



weakly nudged (with a 30-day time scale) to the observed GODAS monthly mean ocean
temperature below a depth of 10 m. Nudging was not applied in the upper 10 m. The SIT
and AGCMs exchange ocean surface fluxes at every time step 48 times a day. AGCMs
were coupled with the SIT in the tropical region between 30°S and 30°N and forced by
prescribed climatological monthly mean SST.

The experiments included three sets of coupled AGCM simulations (ECHAM5-SIT,

CAM5-SIT, and HiRAM-SIT) and standalone AGCM simulations forced by observed
monthly mean OISST (ECHAM5, CAM5, and HiRAM) from 1985 to 2005. The
experiments were designed to evaluate the effect of atmosphere–ocean coupling on MJO
simulations. Table 1 presents the model and experiment details. Due to the computation
limitation of a high-resolution forecast model, the CWBGFS-SIT was only run for 3 years
to test the coupling effect. Thus, its results were evaluated but not compared with those
of the other three models.

The analysis focused on the boreal cool season (November–April) when the

eastward propagation tendency of the MJO is the most prominent. We used the CLIVAR
MJO Working Group diagnostics package (CLIVAR 2009) and a 20–100-day filter to
analyze intraseasonal variability. The MJO phase composites were computed using the
real-time multivariate MJO index (Wheeler and Hendon 2004), which is defined as the
leading pair of principal components of intraseasonal OLR, and 850 and 200 hPa zonal
winds in the tropics.

The vertically integrated MSE budget was diagnosed based on the following

equation:
$$\langle \frac{\partial h}{\partial t} \rangle' = -\langle u \frac{\partial h}{\partial x} \rangle' - \langle v \frac{\partial h}{\partial y} \rangle' - \langle \omega \frac{\partial h}{\partial p} \rangle' + \langle LW \rangle' + \langle SW \rangle' + \langle LH \rangle' + \langle SH \rangle' \qquad (1)$$



where *h* is the MSE ($h = cpT + gz + Lq$); *u* and *v* are the zonal and meridional velocities,
respectively; *ω* is the vertical pressure velocity; *LW* and *SW* are the longwave and
shortwave radiation fluxes, respectively; and *LH* and *SH* are the latent and sensible
surface heat fluxes, respectively. The mass-weighted vertical integration from the surface
to 200 hPa is denoted as ⟨·⟩, and intraseasonal anomalies are represented as ⟨·⟩'. All fields
were isolated using a 20–100-day bandpass Lanczos filter (Duchon 1979).

**3 Results**
**3.1 MJO simulations: ECHAM5-SIT, CAM5-SIT, and HiRAM-SIT**
**3.1.1 General structure**

We compared simulated MJO characteristics using three coupled and uncoupled

AGCMs. Figure 1 presents the wavenumber–frequency spectra of simulated 850 hPa
zonal wind (shading) and precipitation (contours). All three uncoupled AGCMs (hereafter
referred to as AGCMs) simulated intraseasonal signals with lower frequency than the
observed and overestimated the westward propagation with periods >80 days (Fig. 1e–g).
ECHAM5 and HiRAM simulated signals of wavenumbers 1–3 instead of the observed
wavenumber 1 in 850 hPa zonal wind. These results indicate that all three AGCMs
simulated stationary fluctuations with low frequency that were not consistent with the
observation. By contrast, coupled AGCMs realistically reproduce the observed spectral
characteristics and strength of the eastward propagation at wavenumbers 1 to 2 in 850-
hPa zonal wind (Fig. 1b–d). Although HiRAM simulated eastward propagation in a wider
frequency spectrum than that observed, the coupled model clearly displays improvements



in the MJO simulation compared with the stationary intraseasonal fluctuation in the
uncoupled simulation. Hovmöller diagrams presented in Fig. 2 illustrate the temporal
evolution of 850 hPa zonal wind and precipitation in the tropics in observation and
simulations. All three models simulated either stationary (CAM5 and HiRAM) or weak
eastward-propagating (ECHAM5) signals in AGCMs, but more realistically simulated
the eastward propagation of the MJO coupled AGCMs, although the propagation in the
ECHAM5-SIT is still slightly slower than that observed. The improvement obtained in
coupled models suggests that active ocean–atmosphere interaction is a crucial factor for
the successful simulation of the MJO.
**3.1.2 Atmospheric and oceanic profiles**

The composite MJO life cycle featuring intraseasonal OLR and 10-m surface wind

anomalies for boreal winter in eight phases following Wheeler and Hendon (2004) is
displayed in Fig. S1–S3. All three coupled AGCMs simulated realistic MJO with
enhanced circulations and propagation tendency compared with the uncoupled AGCMs.
Figure 3 shows the temporal evolution of vertical heating profiles (averaged over 10°S–
EQ, 120°E–150°E) in eight MJO phases. Observed heating profiles, calculated following
the definition of the apparent heat source (Q1) (Yanai et al. 1973), exhibit diabatic heating
with a maximum near 500 hPa in phases 4 and 5 and in the lower troposphere in earlier
phases. This reflects the development from shallow to deep heating during the
development stage of the convective phase in an MJO. Both ECHAM and HiRAM exhibit
stronger heating in coupled simulations than in uncoupled simulations, whereas the
difference is not evident in CAM5. The vertical structures of the apparent moisture sink
(Q2; contours) associated with the MJO demonstrate a similar convection development.





MJO analysis in phase 4 when deep convection is the strongest over the Maritime
Continent demonstrates the large-scale zonally overturning circulation coupling with the
convection (Fig. 4). The positive heating region in the coupled experiment is significantly
enlarged, deepened, and westward-tilted with increasing height compared with those in
the uncoupled experiment. Correspondingly, the convective-circulation envelope of the
MJO is thicker and longitudinally wider in coupled experiments. The strong convection
is associated with much enhanced low-level moisture convergence (green contours).
Furthermore, the area of positive rainfall anomaly in the coupled experiment becomes
larger, and the sea level pressure anomaly is meridionally more confined, exhibiting the
characteristics of intensified Kelvin wave-like perturbations to the east of the deep
convection. This enhancement of low-level moisture convergence is consistent with the
frictional wave–conditional instability of the second kind mechanism (Wang and Rui
1990; Kang et al. 2013).
In addition to the atmospheric structure, the SST (Fig. S4) and vertical profile of
ocean temperature (Fig. S5) examined are presented in Fig. S5. The observed SST
variation in MJO variability is well reproduced in all three coupled models (Fig. S4). The
warm SST leads the main MJO convection by approximately 5–10 days and is followed
by the cold SST approximately 5–10 days later (Flatau et al. 1997; DeMott et al. 2015;
Tseng et al. 2015). Moreover, the observed amplitude fluctuation (approximately 0.5° to
1°) is realistically simulated. Observed ocean temperature profiles, characterized by the
warm layer, along the equator from the Indian Ocean to the western Pacific are well
simulated in the three coupled models (Fig. S5). Simulated temperature anomalies are
larger in ECHAM5-SIT than in CAM5-SIT and HiRAM-SIT. These results consistently
obtained in all three coupled models support the conclusion of Tseng et al. (2015) that





resolving fine vertical resolution in the upper ocean improves the simulation of warm
layer and MJO propagation and variability. The effect of atmosphere–ocean coupling on
the MJO is independent of AGCMs with different configurations and atmospheric
physical parameterizations. Modifying atmospheric physical parameterizations has been
shown to improve MJO simulation to some extent (Wang et al. 2021), and the results
could be model dependent. Our results demonstrate that the impact of atmosphere–ocean
coupling independent of physical schemes seems to be a more fundamental approach.
**3.1.3 Performance comparison**
To summarize improvements resulting from coupling, simulation was evaluated
(Fig. 5). Figure 5a presents the scatter plot of the power ratio of east–west propagating
waves (X-axis) versus the pattern correlation between the simulated and observed
precipitation anomaly in Hovmöller diagrams (Fig. 2) (Y-axis). The east:west ratio was
calculated by dividing eastward-propagating power by westward-propagating power of
850 hPa zonal wind summed over wavenumbers 1–2 and a period of 30–80 days.
Compared with the observation, coupled simulations (marked by circles) exhibit better
simulation than uncoupled simulations (marked by asterisks). A comparison of combined
explained variance by using RMM1 and RMM2 (Fig. 5b) based on Wheeler and Hendon
(2004) shows marked increases after coupling. A comparison of the coupled and
uncoupled simulations demonstrates that coupling is an essential factor for realistic MJO
simulations.
**3.2 Mechanism discussion**
Here, the MSE budget was applied to diagnose the moisture budget associated with
the MJO. Figure 6 presents a Hovmöller diagram of MSE tendency averaged by 10°S–





EQ overlaying precipitation anomalies. MSE tendency changes in quadrature with
precipitation anomaly with positive (negative) MSE tendency, leading (lagging) major
convection by approximately one to two phases (DeMott et al. 2015; DeMott et al. 2016;
DeMott et al. 2019). Coupled models simulate stronger eastward propagation in both
MSE tendency and precipitation anomalies. Stronger MSE tendencies in coupled
simulations are seen in ECHAM5 and HiRAM but are less clear in CAM5. The
differences between coupled and uncoupled simulations are presented in Fig. 6d, g, j. One
notable feature is the positive (negative) MSE tendency preceding positive (negative)
precipitation anomaly and preconditions an environment for eastward propagation of
active (inactive) convection and associated circulation. We diagnosed the relative
contribution of each term in Equation 1 to the MSE tendency with the focus on the MC,
where the largest positive MSE tendency and precipitation anomaly were found.
**3.2.1 Preconditioning phase**
Following the peak MSE tendency over the MC (120°E–150°E) during phase 2 (Fig.
6d, g, j), values of each term contributing to the column-integrated MSE tendency in
Equation 1 during phase 2 preceding the deep convection over the MC area (10°S–EQ,
120°E–150°E) are displayed in Fig. 7. Vertical advection is the dominant term with the
major compensation from long-wave radiation during phase 2 when convection is still in
the eastern Indian Ocean, as identified by Wang et al. (2017). However, this effect is not
better simulated in the coupled experiments than in the uncoupled experiments in all three
models. Notably, the LH term is consistent between both phases. In all three models, the
coupling reduces the negative MSE tendency. The results indicate that the contribution
comes for the LH in this early phase stage. The LH effect was overlooked in Tseng et al.



(2015) because of the weak MJO variability in coupled simulations. However, this
smaller LH negative became one of the key factors in enhancing the leading MSE
tendency during the MJO preconditioning phases. This suggests that by involving the
coupling process in the preconditioning phase, the surface latent flux bias in AGCMs can
be corrected. In general, coupling improves the simulation of budget. The positive
contribution of vertical advection and negative contribution of LH in MSE tendency is
closer to realistic in the coupled simulations during the initial phase of the MJO.
**3.2.2 Phase of strongest convection over MC**
We compared the spatial distribution of MSE and precipitation in phase 4 when
convection was the strongest in the MC (Fig. 8). In the observation, the main convection
occurs in the MC from 90°E to 150°E. A positive MSE tendency with a maximum near
10°N and 10°S is identified in the east of the MJO convection centered near the equator.
Conversely, a negative integrated MSE tendency is found in the west of the MJO
convection, and the meridionally confined structure near the equator seems to exhibit the
characteristics of the equatorial Kelvin wave embedded in the MJO. Clearly, coupled
models outperform uncoupled models in reproducing these signals. To quantify the
contribution of coupling to the improvement, we follow Jiang et al. (2018) to project all
MSE terms to the observations (Fig. 9). The dominant contribution of horizontal
advection to the MSE tendency in observation (Fig. 9a) is well simulated in the coupled
simulations but not in uncoupled simulations by ECHAM5 and CAM5 (Fig. 9b, c).
Although a similar dominant effect is noted in both simulation types in HiRAM, it is more
enhanced in the coupled simulation (Fig. 9d). The horizontal advection term is further
decomposed into zonal and meridional components (Fig. 9e–h); both components have a
positive contribution, but the meridional component has a larger amplitude. Uncoupled
ECHAM5 and CAM5 simulate unrealistic features: positive contribution from zonal





advection but negative contribution from meridional advection. By contrast, coupled
models well simulate the dominance of meridional advection. In HiRAM, the uncoupled
model simulates almost equally positive contributions from both terms, but the coupled
model is able to simulate the larger contribution from meridional advection. We further
decompose the meridional advection to assess the relative contributions of intraseasonal
perturbation and the mean state. Consistent with the observations (Fig. 9i), the meridional
advection by intraseasonal flow ($-v'\frac{\partial \bar{h}}{\partial y}$) is the main contribution to improve the
simulations in the coupled models (Fig. 9j–l). Our results are consistent with those of
Jiang et al. (2018). To evaluate the relative contribution of intraseasonal circulation and
background moisture, changes in $\Delta(-v'\frac{\partial \bar{h}}{\partial y})$ at phase 4 were further diagnosed. Overbar
denotes that the time mean and prime represents intraseasonal anomaly. Changes in the
MJO meridional advection term for coupled experiments relative to uncoupled can be
written as follows:
$\Delta\left(-v'\frac{\partial \bar{m}}{\partial y}\right) = -\Delta v'\left(\frac{\partial \bar{m}}{\partial y}\right)_{uncoupled} - (v')_{uncoupled}\Delta\left(\frac{\partial \bar{m}}{\partial y}\right) - \Delta v'\Delta\left(\frac{\partial \bar{m}}{\partial y}\right)$ (2)
$(a)$ $(b)$ $(c)$
where $\Delta$ represents the coupled–uncoupled change. The terms a–c are presented as bar
charts in Fig. 10. Notably, the change of the intraseasonal circulation in the meridional
circulation is the dominant factor in coupled simulations relative to uncoupled
experiments. The results confirm that the dominance of dynamic influence over
thermodynamic response to atmosphere–ocean coupling is the key process leading to an
improvement in MJO simulations.

**3.3 Discussion: mean state and intraseasonal variance**





We examined the simulated mean state, which is a major issue affecting MJO
simulations (Inness et al. 2003; Watterson and Syktus 2007; Kim et al. 2009; Kim et al.
2011; Kim et al. 2014; Jiang et al. 2018; Jiang et al. 2020). The three models exhibited
different tropical SST responses to coupling (Fig. S6e). Over the warm pool area, both
CAM-SIT and HiRAM-SIT underestimate the SST, whereas ECHAM5-SIT
overestimates the SST. Warm SST bias in the eastern tropical Pacific was simulated in
the three models because of the lack of oceanic circulation in the SIT. The simulated zonal
wind in the three models (Fig. S6b–d) demonstrated different responses to coupling.
Figure S6c, d presents the 850 hPa zonal wind differences between coupled and
uncoupled models (shading) and the total field in uncoupled models (contours). Figure
S6f–h shows the 10°S–EQ averaged 850 hPa zonal in both coupled and uncoupled models.
In ECHAM5-SIT, the westerly wind is slightly enhanced in the eastern Indian Ocean but
decreases in the western Indian Ocean and western Pacific. In CAM5-SIT, westerly wind
reduces in the Indian Ocean but enhances over the western Pacific. The HiRAM-SIT has
similar changes as in ECHAM5-SIT, with decreases over the Maritime Continent area
but increases in the western Indian Ocean and Pacific. In general, the three models
disagree in the changes in zonal wind mean state in response to coupling.
The mean moisture changes are substantially enhanced over the tropical areas in
ECHAM5 after coupling (Fig. S7b, e). However, in both CAM5 and HiRAM, no clear
change was observed to the south, but strong drying was observed to the north of equator
(Fig. S7c, d, f, g). The only common feature among the three models that is enhanced in
the coupled simulations is the meridional gradient of mean moisture. This is consistent
with many previous studies (Kim et al. 2014; Jiang et al. 2018; Ahn et al. 2020). Our
budget analysis indicated that the meridional transport by the intraseasonal meridional



circulation is the dominant term, and the meridional gradient of mean moisture is the
secondary effect in enhancing MJO simulations by coupling. The mean precipitation
changes are more consistent among the three models after coupling (Fig. S8). One of the
major changes is the southward shift of the major precipitation zone, resulting in
precipitation increases over the regions south of the equator except in the Maritime
Continent. Similarly, the precipitation intraseasonal variance (20–100 days filtered)
markedly enhances in these regions (Fig. S9). The ECHAM5-SIT exhibits a relatively
minor increase over the western Maritime Continent. By contrast, the HiRAM-SIT
exhibits the strongest enhancement, particularly in the Indian Ocean. In general, all three
coupled models enhance the intraseasonal signals over the tropics with discrepancies in
detail. By contrast, the model mean state does not substantially improve after coupling.
Thus, in this study, the mean state is not the main contribution to the enhancement of the
MJO simulation after coupling. Instead, coupling leading to rigorous atmosphere–ocean
interaction is likely the reason for the improvement of MJO simulation.

**3.4 The forecast model: CWBGFS**
CWBGFS and CWBGFS-SIT were compared for only 3 years. Figure 11
demonstrates the wave number–frequency spectra and the 10°S–10°N averaged lag–
longitude diagrams of CWBGFS between coupled and uncoupled versions. The spectra
of CWBGFS-SIT suggest better simulation (Fig. 11a, b) in relation to better propagation
across the MC (Fig. 11c, d). Although we did not examine the mechanisms in detail, our
results demonstrate that MJO forecast skills could be improved by considering the
coupling effect in the forecast model.





## 4 Discussion

This study used a one-column TKE-type ocean mixed-layer model SIT coupled with AGCMs to improve MJO simulation. SIT is designed to have fine layers near the surface and can simulate their warm layer, cool skin, and diurnal fluctuations. This refined discretization under the ocean surface in SIT provides improved SST simulation and, thus, realistic air–sea interaction. Coupling SIT with ECHAM5, CAM5, and HiRAM significantly improves the MJO simulation in the three AGCMs compared with that in prescribed SST-driven AGCMs. The vertical cross section indicates that the strengthened low-level convergence during the preconditioning phase is better simulated in the coupled experiment. Furthermore, the phase variation and amplitude of the SST and ocean temperature under the surface can be realistically simulated. Our results reveal that the MJO can be realistically simulated in terms of strength, period, and propagation speed by increasing the vertical resolution of the one-column ocean model to better resolve the upper-ocean warm layer.

The MSE budget analysis revealed that the coupling effects during the earlier phases and mature phase exhibit different contributions. During the preconditioning phase, the positive contribution of vertical advection and negative contribution of LH in MSE tendency are closer to realistic values in coupled simulations during the initial phase of the MJO. During the mature phase of the strongest convection in the MC, the meridional component of the horizontal advection term is the dominant term to enhance the simulation after coupling. Improved meridional circulation is essential in the coupled simulations that outperformed uncoupled experiments. The results confirm that the dominance of dynamic influence over thermodynamic influence in response to the





atmosphere–ocean coupling is the key process leading to the improvement of MJO
simulations.
In summary, this study suggests two major enhancements of the coupling process.
First, during the preconditioning phase of the MJO over MC, the underestimated surface
LH bias in AGCMs can be corrected. Second, during the strongest convection phase over
MC, the change in intraseasonal circulation in the meridional circulation is the dominant
factor in coupled simulations relative to uncoupled experiments. Although many studies
have indicated the key role played by the mean state, the mean state in our simulations
provides only a secondary contribution to enhancing MJO simulation, with coupling
being the main contributor. For example, zonal wind and precipitation changed
inconsistently among the three models after coupling. Instead, the meridional gradient of
the mean moisture and intraseasonal variance of precipitation have a better relationship
after coupling. Therefore, coupling leading to rigorous atmosphere–ocean interaction, but
not change in mean states, is likely the reason for MJO simulation improvement.
Moreover, coupling SIT with the weather forecast model CWBCFS can improve MJO.
This study supports previous findings (Tseng et al. 2015) that the enhancement of
atmosphere–ocean coupling by considering extremely high vertical resolution in the first
few meters of the ocean model improves MJO simulations, and this improvement is
independent of AGCMs with different configurations and physical parameterization
schemes. Resolving the atmosphere–ocean coupling may be more beneficial than
modifying the atmospheric physical parameterization schemes in GCM.



**Code and data availability.** The model code of CAM5 – SIT, ECHAM5-SIT and
HiRAM-SIT is available at https://doi.org/10.5281/zenodo.5701538,
https://doi.org/10.5281/zenodo.5510795 and https://doi.org/10.5281/zenodo.5701579.
Observational data used in this study include precipitation from Global
Precipitation Climatology Project V1.3 (GPCP, 1° resolution), outgoing longwave
radiation (OLR, 1° resolution), and daily SST (Optimum Interpolated SST, 0.25°
resolution) from the National Oceanic and Atmosphere Administration, and variables
were obtained from the European Centre for Medium-range Weather Forecast
Reanalysis-interim. All model codes and data availability presented here can be obtained
by contacting the first author, Dr. Wan-Ling Tseng (wtseng@gate.sinica.edu.tw).

**Author contributions.** HHH and WLT have responsibility for conceptualization,
including analyzing the data and writing the manuscript. YYL, PHK, BJT, CYT and HCL
developed the model and provided the simulations.

**Competing interests.** The authors declare that they have no conflict of interest.

**Acknowledgments.** This work was supported by the Taiwan Ministry of Science and
Technology under grant numbers MOST 109-2111-M-001-012-MY3, MOST 110-2811-
M-001-633, and MOST 110-2123-M-001-003. We are grateful to the National Center for
High-Performance Computing for providing computer facilities. The Max Planck
Institute for Meteorology provided ECHAM5.4. We sincerely thank the National Center
for Atmospheric Research and their Atmosphere Model Working Group (AMWG) for
release CESM1.2.2. This manuscript was edited by Wallace Academic Editing.





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





|  |  | ECHAM5-SIT | CAM5-SIT | HiRAM-SIT | CWBGFS-SIT |
|---|---|---|---|---|---|
| AGCM |  | ECHAM5 | CAM5 | HiRAM | CWBGFS |
| Horizontal resolution |  | T63(~2°) | 1.9°X2.5° | 1°X1° | T319 |
| BC | SST | OISST | OISST | OISST | OISST |
|  | SIC | OISST | OISST | OISST | OISST |
|  | OT/OS | GODAS | GODAS | GODAS | GODAS |
| Atmosphere vertical resolution |  | L31 | L30 | L32 | L60 |
| Ocean vertical resolution |  | 42 | 42 | 42 | 42 |
| Coupled region |  | 30˚S-30˚N | 30˚S-30˚N | 30˚S-30˚N | 30˚S-30˚N,30˚-40˚, blending interpolated |
| Time |  | 1985-2005 (21 years) |  |  | 2012-2014 (3 years) |

**Table 1.** Detailed information of models and experiments.






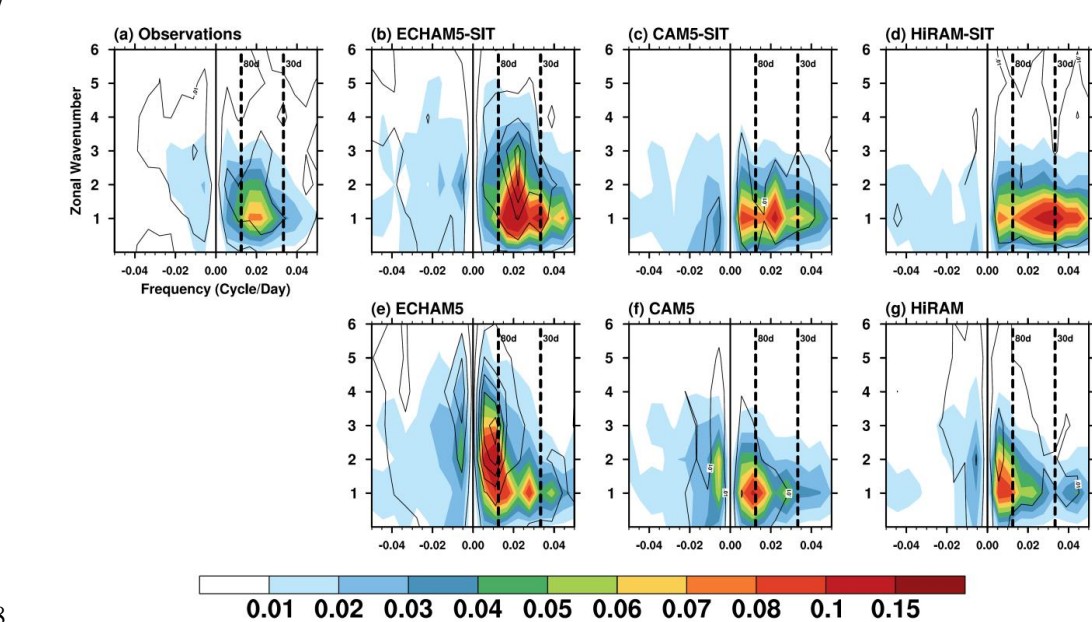


**Figure 1.** Wave number–frequency spectra for equatorial 850 hPa zonal wind (shading)

and precipitation (contours) over 10°S–10°N from (a) observations and simulations by

using the (b–d) coupled and (e–g) uncoupled AGCM.







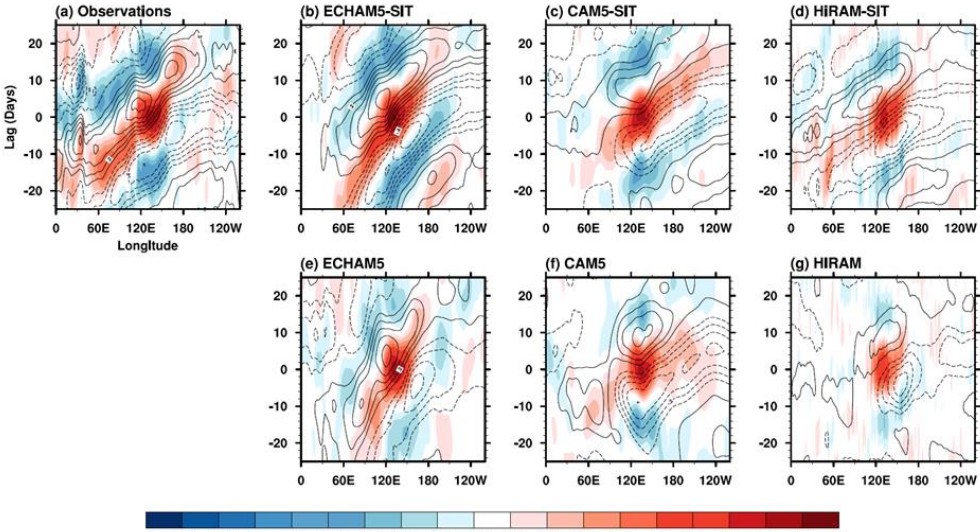


**Figure 2.** The 10°S–10°N averaged lag–longitude diagrams of intraseasonal precipitation

(shading) and 10-m zonal wind (contour) correlated against precipitation at region (10°S–

5°N, 120°E–150°E) from (a) observations and simulations by using the (b–d) coupled

and (e–g) uncoupled AGCM. The contour interval is 0.1.

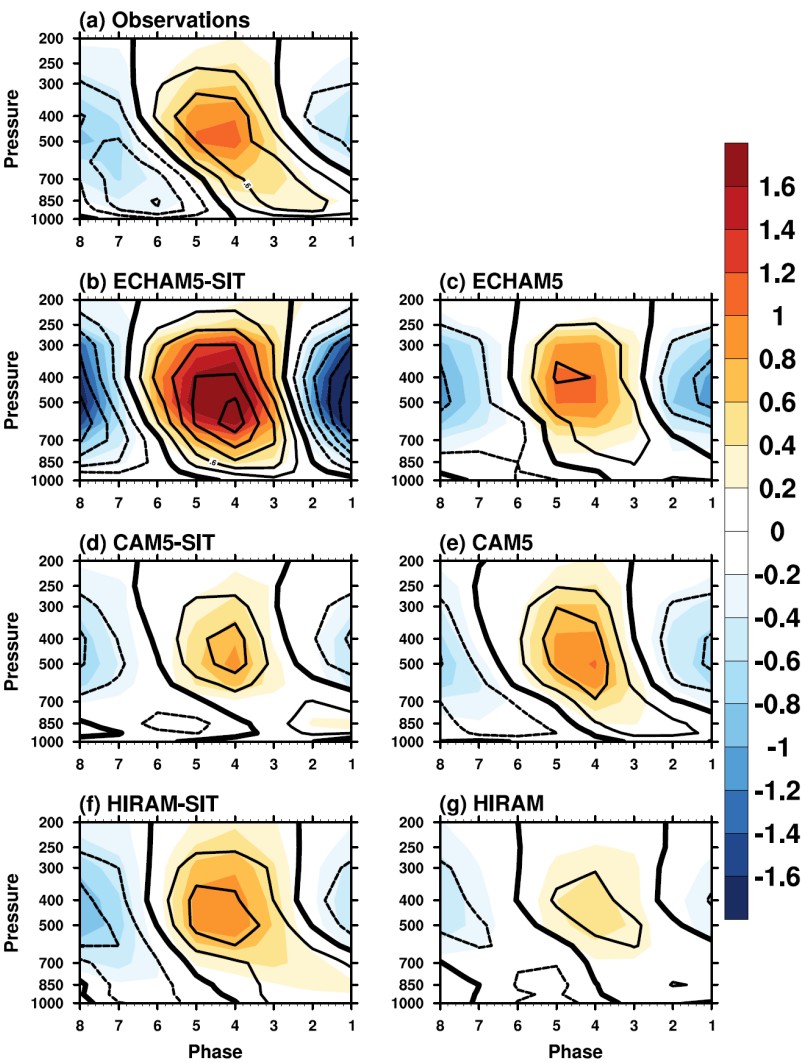

**Figure 3.** Vertical profiles with respect to MJO phases averaged over 10°S–EQ and 120°E–150°E for intraseasonal anomalies (i.e., with 20–100-day filtering) of Q1 (shading; K day$^{-1}$) and Q2 (contours; K day$^{-1}$) from (a) observations and simulations by using the (b–d) coupled and (e–g) uncoupled AGCM.





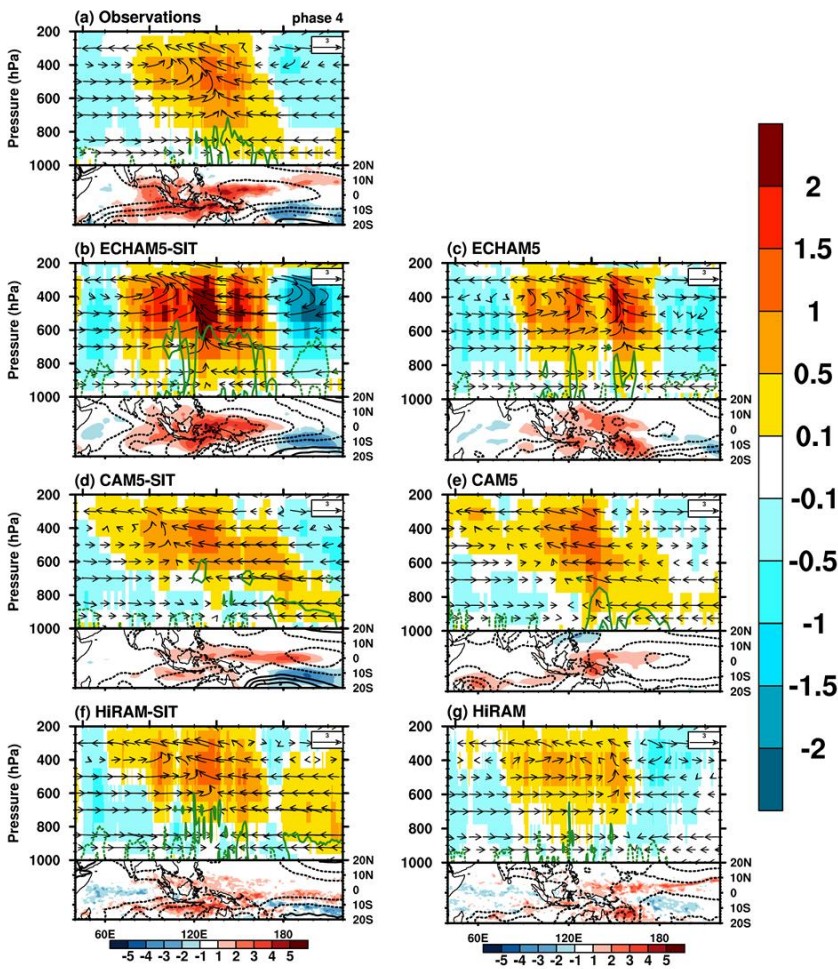

**Figure 4.** Structure of simulated MJO in phase 4. The longitude–height cross-sections

(averaged over 10°S–EQ) of the MJO scaled wind circulation (vector, u: m s$^{-1}$, omega:

10$^{-2}$ Pa s$^{-1}$), Q1 (shading, unit: K day$^{-1}$), and the horizontal moisture convergence (green

contour, unit: 10$^{-6}$ g kg$^{-1}$ s$^{-1}$) from (a) observations and simulations by using the (b–d)

coupled and (e–g) uncoupled AGCM. The contour interval of the moisture convergence

is $8 \times 10^{-6}$ g kg$^{-1}$ s$^{-1}$; solid line is positive. Precipitation (shading, unit: mm day$^{-1}$) and sea



level pressure (contour, unit: hPa). Contour interval of sea level pressure is 30 hPa; dashed

line indicates negative.

656



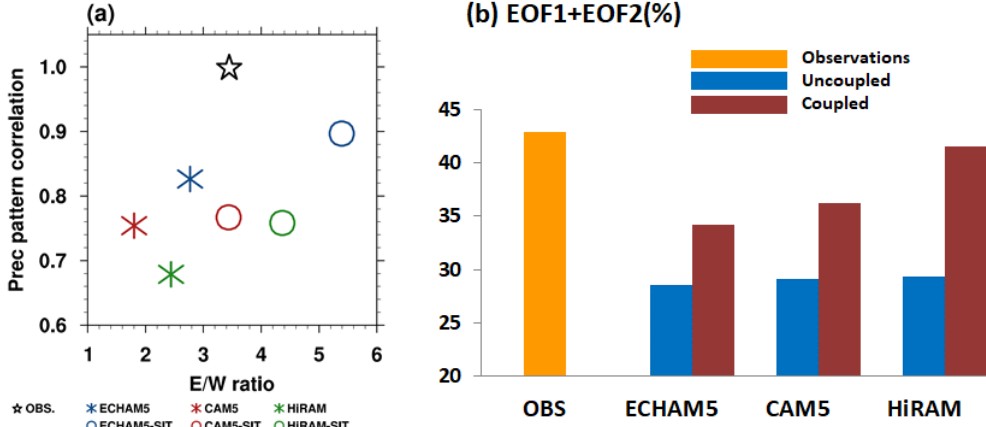

**Figure 5.** Scatter plots of various MJO indices based on observation and experiments

(Table 1). (a) X-axis is the power ratio of east–west propagating waves. The east–west

ratio was calculated through the division of the sum of eastward-propagating power by

the westward-propagating counterpart within wavenumbers 1–3 (1–2 for zonal wind),

period 30–80 days. The Y-axis is the pattern correlation of precipitation eastward

propagation, as shown in Fig. 2. (b) Sum of RMM1 and RMM2 variances based on

Wheeler and Hendon (2004).

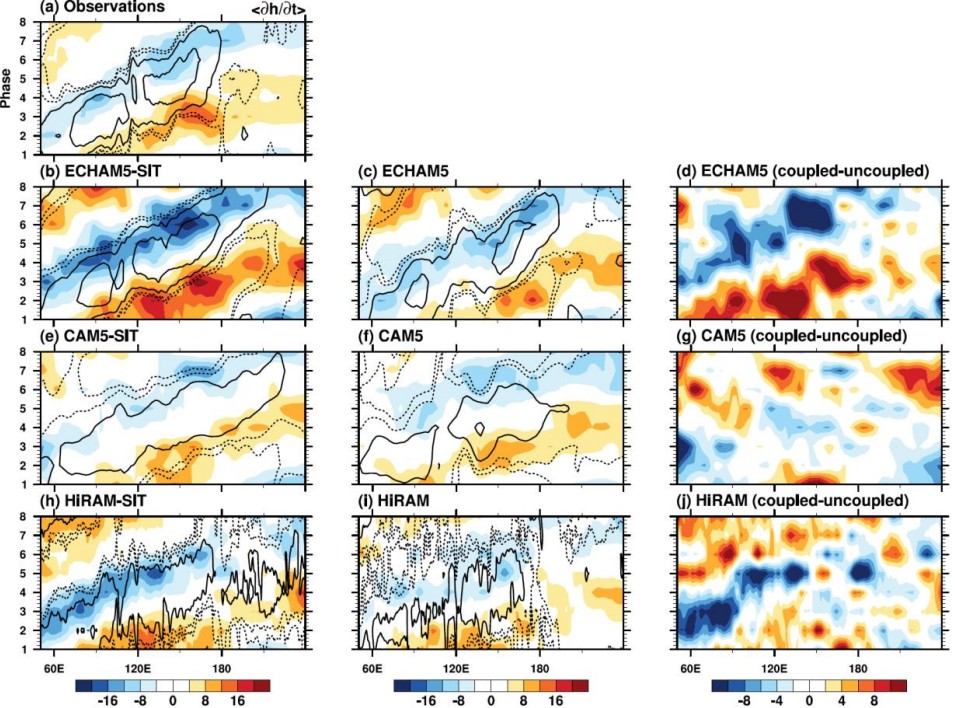

**Figure 6.** The 10°S–EQ averaged Hovmöller diagrams of MSE (shading) and precipitation (contour) composite followed the RMM index from (a) observations and simulations by using the (b, e, j) coupled and (c, f, k) uncoupled AGCM and (d, i, l) their difference. The contour interval is precipitation anomalies.





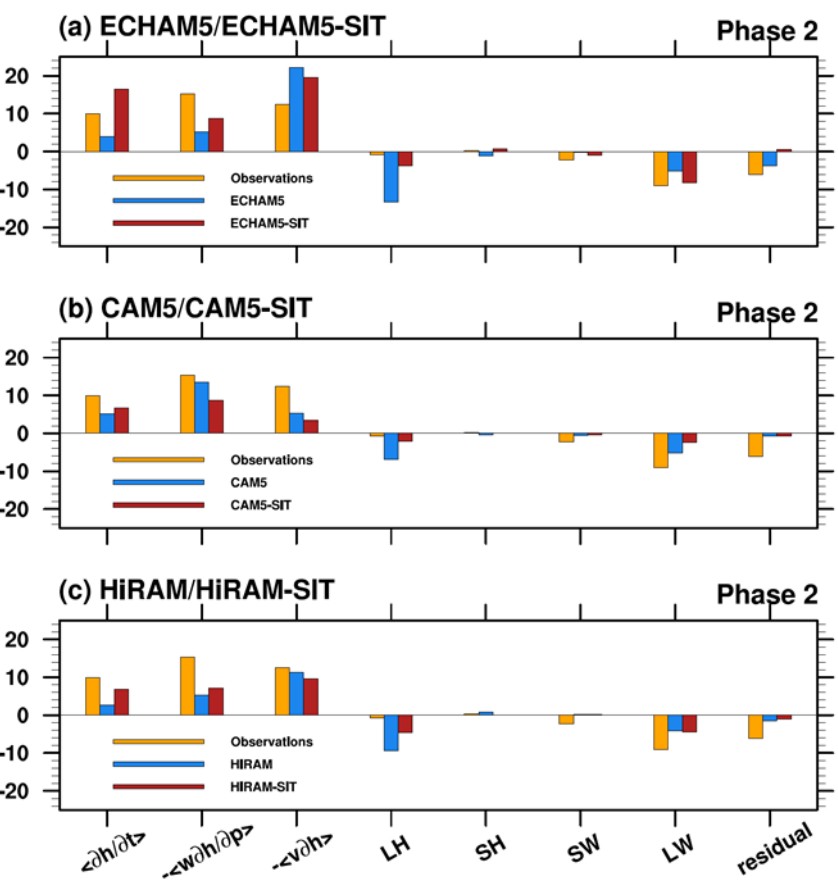

673

**Figure 7.** Model-simulated column-integrated MSE budget terms (J kg$^{-1}$ s$^{-1}$) during

phase 2 of the MJO. Data from the observations, Nordeng scheme simulation, and Tiedtke

scheme simulation are shown in black, red, and blue, respectively. The averaged domain

is 10°S–EQ and 120°–150°E.






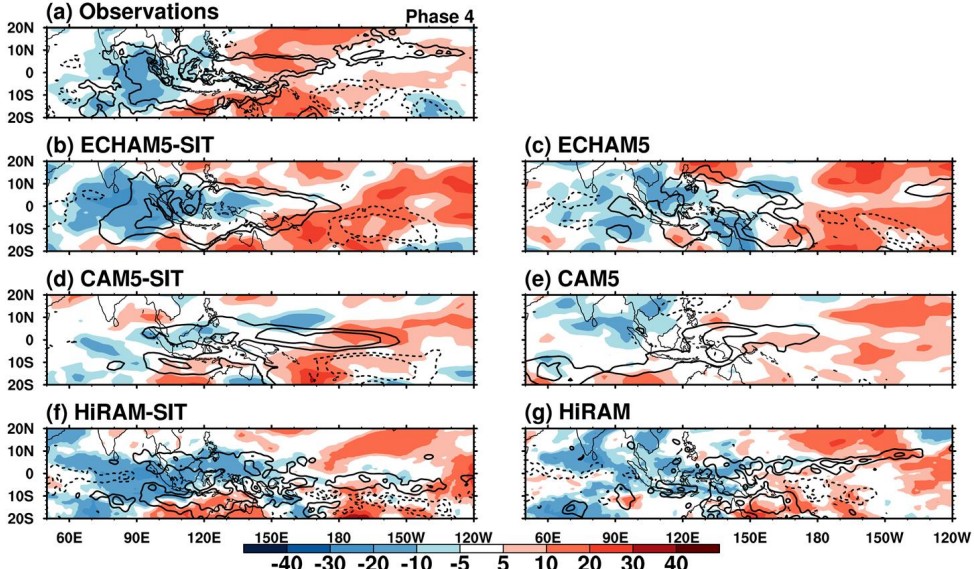


**Figure 8.** Phase 4 of the column-integrated MSE tendency (shading) and precipitation

(contours) based on (a) observation, (b) ECHAM5-SIT, (c) ECHAM5, (d) CAM5-SIT,

(e) CAM5, (g) HiRAM-SIT, and (f) HiRAM. The nine-point local smoothing is applied

in the intraseasonal precipitation variance of HiRAM here (contours only).




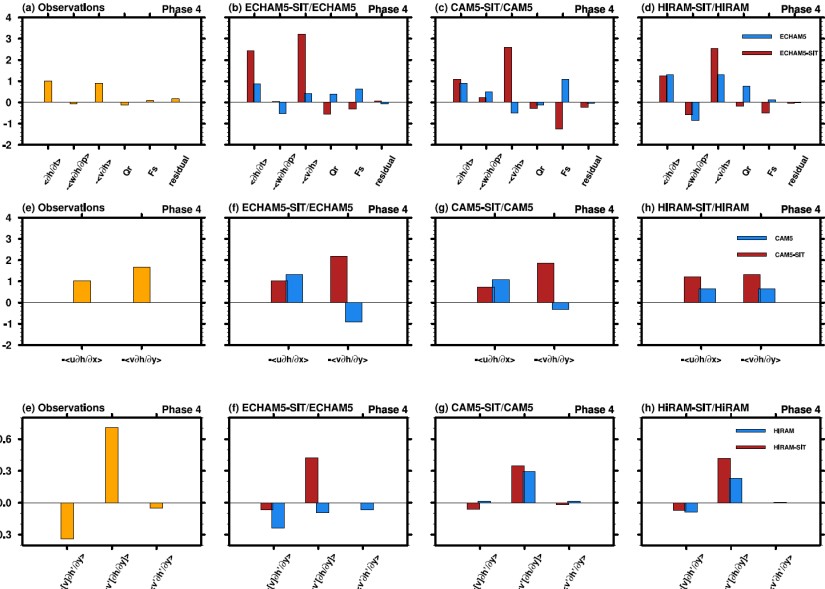


**Figure 9.** (a–d) Relative role of each MSE component of phase 4 through the projection of the spatial pattern of each MSE budget over the MC (domain) onto the total MSE tendency pattern (Fig. 8a). (e–h) Decomposite of the total horizontal MSE advection based on zonal and meridional components. (i–l) Decomposite of the meridional horizontal MSE advection based on the MJO circulation and the mean state of moisture.

695





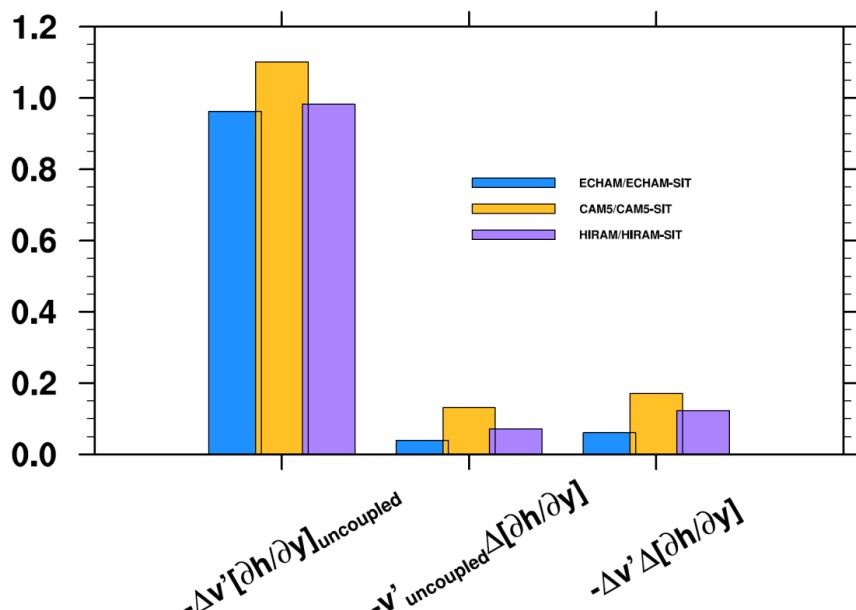

**Figure 10.** Bar chart of relative contribution of intraseasonal convergence and background moisture between the coupled and uncoupled change in MJO phase 4.





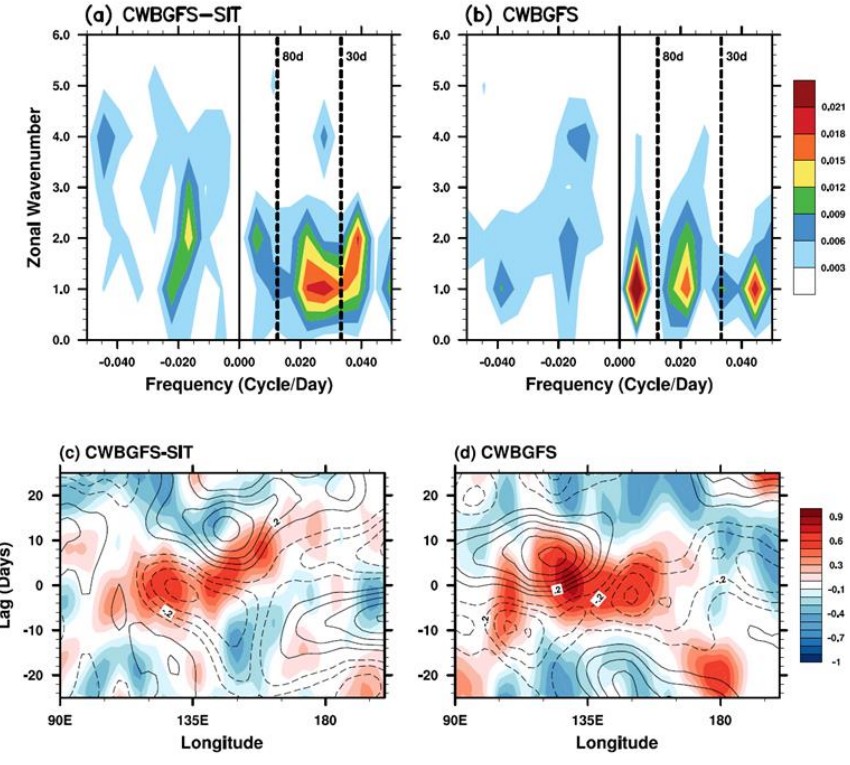

701

**Figure 11.** (a, b) Same as Fig. 1, but in CWBGFS-SIT and CWBGFS. (c, d) Same as Fig.

2, but in CWBGFS-SIT and CWBGFS.

704

705