# Peer review of "Improving Madden-Julian Oscillation Simulation in Atmospheric General"

_Geoscientific Model Development, 2021_

## Author Comment (AC1)

Review paper "Improving Madden–Julian Oscillation Simulation in Atmospheric General Circulation Models by Coupling with Snow–Ice–Thermocline One-dimensional Ocean Model".

**[RC1]**

This is an interesting work investigating the improvement of the MJO simulation by coupling the AMIP to the Sea-Ice-Thermocline single-column model. And the most important is the fine resolution of the upper oceanic temperature could play such an important role. From the MSE analysis, it is apparent to observe the prominence of the latent heat, which has been underestimated in AMIP simulation. However, it is not sure if that is also the case in the coupled models. The authors mentioned the diurnal warm and cold skin; however, it is not addressed well in the paper. If the authors could provide more explanation or references on it is suggested. Figure S5 is a very interesting plot. Although all experiments use the same SIT module, the temperature penetration depth seems very different in between models. The depth is different, but the stronger variance is shown in the ECHAM5-SIT experiment. To have the storage when running the HIRAM-SIT is understandable, but the magnitude of the 3m at the 1.5S, 90E seems much weaker at the 9m, and the centre seems shifted. What will cause that different pentation? I hope the authors could provide a little explanation, and it might be useful information for the global coupled model teams. I am happy with this version of the article and agree the article meets the standard of the GMD journal. However, an extra description of the oceanic dynamics and the labelling adjustment in Figure 5S will be more appreciated.

**Reply to reviewer:**
Thank you for the sincere comments regarding our manuscript, as well as taking the time to provide several suggestions for the improvement. In particular, the comment about the oceanic response is less addressed in our previous manuscript. Here, our point-by-point responses to each of the individual comments are outlined below.

1. From the MSE analysis, it is apparent to observe the prominence of the latent heat, which has been underestimated in AMIP simulation. However, it is not sure if that is also the case in the coupled models.

   In the coupled model, the synchronous ocean surface temperature is varying associated with the MJO variability. Therefore, the LH flux biases are smaller than AMIP simulations. It is the main contribution of the coupling during the MJO

preconditioning phase. Previous studies have identified that coupled models tend to perform better simulations (Demott et al., 2019; Jiang et al., 2015).

2. The authors mentioned the diurnal warm and cold skin; however, it is not addressed well in the paper. If the authors could provide more explanation or references on it is suggested.

Thank you for the reminder. More details and references are added.

"Cool skin is a very thin layer that has a direct contact with the atmosphere and warm layer is the warmer sea water immediately below the cool skin in the top few meters of the ocean. They fluctuate diurnally in response to atmospheric forcing. SIT with high vertical resolution realistically simulates the warm-layer (within top 10 m) and cool-skin (the top layer with 0.001 m thickness), and improve the simulation of upper ocean temperature (Tsuang et al., 2009; Tu and Tsuang, 2005). The model has been verified at a tropical ocean site (Tu and Tsuang, 2005), in the South China Sea (Lan et al., 2010), and in the Caspian Sea (Tsuang et al., 2001). The melt and formation of snow and ice above a water column has been introduced (Tsuang et al., 2001)."

3. Figure S5 is a very interesting plot. Although all experiments use the same SIT module, the temperature penetration depth seems very different in between models. The depth is different, but the stronger variance is shown in the ECHAM5-SIT experiment. To have the storage when running the HIRAM-SIT is understandable, but the magnitude of the 3m at the 1.5S, 90E seems much weaker at the 9m, and the centre seems shifted. What will cause that different pattern? I hope the authors could provide a little explanation, and it might be useful information for the global coupled model teams.

Thank you for the suggestion. We have moved Fig. S5 to main figure (Fig. 6). The differences between models are likely due to the different atmospheric model configurations, because they were coupled to the same 1-D ocean model. Since the atmosphere is the main driver to extract heat form the ocean, different responses of atmospheric models likely have different effects on SST. In our study, we clearly demonstrate that coupling improves the simulations in three AGCMs with very different configurations and parameterization schemes. The cause of quantitative differences in subsurface temperature between models is not explored in this study. Further detailed analysis is needed to pinpoint. We have added this discussion.

DeMott, C. A., Klingaman, N. P., Tseng, W. L., Burt, M. A., Gao, Y., and Randall, D. A.: The convection connection: How ocean feedbacks affect tropical mean moisture and MJO propagation, Journal of Geophysical Research: Atmospheres, 124, 11910-11931, 2019.

Jiang, X., Waliser, D. E., Xavier, P. K., Petch, J., Klingaman, N. P., Woolnough, S. J., Guan, B., Bellon, G., Crueger, T., DeMott, C., Hannay, C., Lin, H., Hu, W., Kim, D., Lappen, C.-L., Lu, M.-M., Ma, H.-Y., Miyakawa, T., Ridout, J. A., Schubert, S. D., Scinocca, J., Seo, K.-H., Shindo, E., Song, X., Stan, C., Tseng, W.-L., Wang, W., Wu, T., Wyser, K., Wu, X., Zhang, G. J., and Zhu, H.: Vertical structure and diabatic processes of the Madden-Julian Oscillation: Exploring Key Model Physics in Cimate Simulations, J. Geophys. Res. Atmos. (under revision), 2015.

Lan, Y.-Y., Tsuang, B.-J., Tu, C.-Y., Wu, T.-Y., Chen, Y.-L., and Hsieh, C.-I.: Observation and simulation of meteorology and surface energy components over the South China Sea in summers of 2004 and 2006, Terrestrial, Atmospheric and Oceanic Sciences, 21, 325-342, 2010.

Tsuang, B.-J., Tu, C.-Y., and Arpe, K.: Lake parameterization for climate models., Max-Planck-Institute for Meteorology Rept, 316, 72pp, 2001.

Tsuang, B.-J., Tu, C.-Y., Tsai, J.-L., Dracup, J. A., Arpe, K., and Meyers, T.: A more accurate scheme for calculating Earths-skin temperature, Climate Dynamics, 32, 251-272, 2009.

Tu, C.-Y. and Tsuang, B.-J.: Cool-skin simulation by a one-column ocean model, Geophysical research letters, 32, 2005.

---

## Author Comment (AC2)

Review paper "Improving Madden–Julian Oscillation Simulation in Atmospheric General Circulation Models by Coupling with Snow–Ice–Thermocline One-dimensional Ocean Model".

**[RC2]**

Thank you for the sincere comments regarding our manuscript, as well as taking the time to provide suggestions for the improvement. I am grateful to have your review that help us improve the manuscript. Our point-by-point responses to each comment are listed below.

This paper incorporates a one-dimensional ocean mixed layer model into three atmospheric models, ECHAM5, CAM5 and HiRAM. Specifically, the Madden-Julian Oscillation (MJO) is significantly improved in these three coupled models due to a more realistic simulation of SST variation. The coupled simulations can correct the surface latent heat flux biases during the preconditioned MJO phase over Maritime Continent (MC). The change of meridional circulation during the strong convection phase also control the improvement. In general, this manuscript clearly shows the atmospheric dynamics associated with the enhancement of MJO regardless of model configurations/physics. The budget analysis also details the relative contribution. However, the fundamental driver from the coupled air-sea interaction process which changes the boundary layer through the SST update is still unclear. The authors may have to comment on this further. Finally, the English usage needs further improvement. Careful proofread by a native English writer is required. This paper is appropriate to be published in GMD after considering the following comments.

1. Throughout the manuscript (including the abstract), the use of CWBCFS is mentioned several times. However, we do not see the results until Fig. 11 (section 3.4). The description is also very minimal (one paragraph). Unless more discussion is included, I suggest to remove all discussion about this model results which cannot add any new information in this study.

   Thank you for the suggestion. The part of CWBGFS is removed.

2. Introduction: line 56-59, what's the meaning of this sentence? "MJO and oceanic wave are also suggested"? What? Do you want to say they are related? This sentence has to be rewritten.

The sentence is rewritten as "Besides, oceanic wave dynamics are suggested to be associated with MJO, for example, zonal wind stress anomalies driven by the MJO force eastward-propagating oceanic equatorial Kelvin wave (Hendon et al., 1998; Webber et al., 2010)"

3. Line 66, suggest to remove "evaluating the mechanism of ocean-atmosphere coupling" since the following description is to discuss the mechanism of ocean-atmosphere coupling already.

Thank you for the suggestion. It is removed.

4. Line 73-79, this sentence is unclear, particularly after "Such as". The whole sentence needs to be rewritten.

The sentence has been rewritten.

5. Line 96-98, this sentence does not have a verb.

The part of CWBGFS-SIT is removed.

6. Section 2: I suggest to separate into two subsections. "2.1 Observation and atmospheric/oceanic data" and "2.2 Model experiments" to better clarify the information.

Thank you for the suggestion. The section is now separated to "2.1 Observation and atmospheric/oceanic data", "2.2 Model experiments", and "2.3 Methodology".

7. Line 103-Line 113. This paragraph describes the observational results used here. However, it is not easy to read. Also some information is unclear. What variables are used from ERA-interim since ERA-interim also has precipitation and outgoing longwave radiation? Also, the time periods used look different. Please clarify. What about the oceanic GODAS forecast and TOA array data? What time periods do you use? I suggest the authors to systematically list different datasets. True observation and model data should be clearly separated. Don't mix them all together.

It is revised as your suggestion. The true observation and reanalysis is separated.

8. The only difference between the coupled and uncoupled simulation is the update of SST. The uncoupled simulation specified the SST, however, the coupled

version updated the SST at every time step. Is this correct? Does the coupled simulation feed other variables back to the atmospheric component?

Yes, it is correct. Although the SIT only provides SST to atmospheric model, the updated SST will then change latent and sensible heat fluxes.

9. Line 115-116, "variations in the SST and upper-ocean temperature, including the" change to "upper-ocean temperature, including the SST, " If your cool skin temperature is SST, you can skip "the SST".

Cool skin is the second layer below SST. The depth is 0.5 mm. There is about 0.1-0.2K difference from SST. Therefore, we would like to keep the SST here.

10. Line 119, remove "a".

Thank you for the reminder.

11. Line 123-127, the resolution used in CAM5 and HiRAM need to be described. Also, the boundary layer schemes used in these models should be briefly described to comment on the different boundary layer schemes used here in the coupling.

It has been revised accordingly.

12. Line 127-130, remove this model description since it doesn't add any new information while no results are presented until Fig. 11.

It is removed.

13. Line 133, is this 0.05mm your finest resolution at the top? Does the resolution increase with depth? Also, you have 12 layers in the top 10 m. How many are within the top 1 m so resolve the diurnal warm layer?

We have added the detail description on model vertical resolution. "In this study, we applied 42 vertical layers in SIT, with 12 layers in the upper 10 m at surface, 0.05mm, 1 m, 2 m, 3 m, 4 m, 5 m, 6 m, 7 m, 8 m, 9 m, and 10 m." Fig. R1 is the Diagram showing the vertical grid from Lan et al. (2021).

[Figure]

**C–30NS**

grid center
(U, V, T, S)

grid edge
(flux)

0 m
-0.0005 m

0 m
-0.001 m
-1.5 m

-10.0 m

-10.5 m

-16.84 m

-23.01 m

-29.51 m

-36.34 m

-43.56 m

-51.18 m

-59.25 m

-67.80 m

-76.88 m

-83.53 m

-96.80 m

-107.75 m

Figure R1. Diagram showing the vertical grid within 107.8 m in SIT. Fig. S1 from Lan et al. (2021)

Line 137, to my understanding, the top layer of GODAS is 10m. Do you mean you do not nudge the first top value (i.e., SST) but the values below. So you want to mimic the observed SST (from 1985 to 2005) but not the sea surface dynamic, right? I suggest to include a new plot showing this upper layer feature comparing to the OISST. This may be an important plot to show the major forcing difference on the atmospheric model.

Yes, exactly. Nudging was not applied in the upper 10 m. Fig R2 demonstrates both the observed (TAO array) and simulated climatology of diurnal SST variation. The SIT model can realistically simulate the diurnal SST cycle. Fig R3 demonstrates the diurnal temperature variation in the upper ocean warm layer at 1.5°S, 90°E. Both cool skin and warm layer can be well reproduced.

[Figure]

Figure R2. Observed (TAO) and simulated (ECHAM5-SIT) diurnal cycle of SST at two sites (1.5˚S, 90˚E) and (0˚N, 147˚E). The unit is ˚C. Black curve is the observed and red curve is simulated value.

[Figure]

Figure R3. The climatology diurnal temperature profile at 1.5°S, 90°E. The circles denote as SST. Notice the second layer 0.005m which is not linear in the y-axis. Diurnal fluctuations of warm layer and cool skin are clearly seen.

14. Line 138, if this is the case, all atmospheric models use the same time step? I believe these three models have different resolutions. So do you control the time step on purpose?

Thank you for reminding the mistake. The time step varies between models, i.e., 720 seconds in ECHAM-SIT, 1800 seconds in CAM5-SIT, and 900s second in HiRAM-SIT.

15. Line 140, "prescribed climatological monthly mean SST" do you mean "prescribed monthly mean OISST"? If so, it is better to clarify this.

It is corrected to "prescribed monthly mean OISST".

16. Line 145-148, I suggest to remove the CWBCFS description and comment on this at the summary and discussion section.

It is removed.

17. Figure 1: please include units in the caption or on the figure.

It is revised.

18. Figure 2: please include the units in the caption or on the figure.

Correlations are shown in Figure 2. There are no units.

19. Line 191-194, are Figures S1-S3 very important plots? If so, why they are on the supplementary figures? If not, why do the author discuss them at the beginning? What's the purpose of putting this sentence?

They are important figures. We have put them back in the main text. Previously, we leave them in the supplement to save space.

20. Line 200, change "MJO" to "MJO event".

It is revised.

21. Figure 3: what's the purpose of showing this Figure? Do you want to imply the heat sources are not the key for the MJO development (because CAM5 v.s. CAM5-SIT does not have the corresponding change)?

Figure 3 has been removed.

22. Line 213: How can you justify this is an intensified Kelvin wave-like perturbation? Can you identify the wave propagation or forcing?

The SLP and wind anomalies in bottom panel of Fig. 4 demonstrates the Kelvin wave like structure, e.g., maximum values at the equator and rapid decreases away from the equator. We added the plot of wavenumber-frequency spectra (Fig. R4). It shows the enhancement of the Kelvin wave in the coupled simulation. We have also added this discussion and figure in the main manuscript.

"The enhancement of Kelvin wave can be observed in symmetric wavenumber-frequency spectra (Fig. 5). The spectra between 0 to $0.35 \text{day}^{-1}$ is presented here to highlight the MJO and equatorial Kelvin waves. The coherence at wavenumbers 2–4 for the 10–20-day period is all simulated stronger in coupled models than uncoupled ones."

[Figure]

Figure R4. Wavenumber-frequency spectra of 10°N–10°S-averaged 850-hPa zonal wind. Units: m²s⁻².

23. It is very interesting to see this large difference occurs just above the MC. This region includes both ocean and a large area of land. Particularly, the ocean is very shallow in general. Can you comment on this topography feature on the large impact of the coupling?

The MC is the main region where the MJO convection occurs. Our previous study (Tseng et. al., 2017) demonstrated the contribution of both orography and land–sea contrast contributes markedly to the MJO intensity and propagation. As for the coupling effect, Tseng et al. (2014) showed that the coupling enhances the low-level moistening that preconditions the main convection in an MJO. The moistening process is enhanced through enhanced convergence of moisture flux originating in the surrounding oceans, instead of from local evaporation in the MC. It follows that the coupling in the vast ocean, not just those inside the MC, contributes to the improved simulation of the MJO. Our companion manuscript (Lan and Hsu 2022) that is currently under review in GMD demonstrates that coupling in both the Indian Ocean and the whole tropical Pacific benefits markedly to the improved simulation of the MJO (Lan et al., 2021). In addition, the other manuscript that we are preparing demonstrates the deep ocean bathymetry surrounding the MC islands (Fig. R5). Even in the shallow ocean part such as the Java Sea is deeper than 100 meter and is still characterized by the warm layer that fluctuates in top few meters. In conclusion, the shallowness of the MC oceans does not prevent the coupling effect from impacting the MJO.

[Figure]

**Figure R5.** The cross session of lag regression -5 days at 5°S.  (a) Q1 (shading; k day$^{-1}$) and circulation (u; m s$^{-1}$ and omega; pa s$^{-1}$) in the upper panel. SST (red; °C), LH (blue; w m$^{-2}$) and surface net heat flux (black; w m$^{-2}$) in the middle panel. Potential density (shading; kg m$^{-3}$) and ocean circulation (OU; cm s$^{-1}$ and ocean vertical velocity; 10$^{-5}$ cm s$^{-1}$) in the bottom panel.

Figure S5 seems to be a very important plot for the SIT to resolve the upper 10m ocean. However, the warm layer change look different among these three coupled models. The only consistence I can tell is the SST at different phases (which are used by the atmospheric models). Can you also include the SST at different phases used to force the uncoupled model for the comparison? This may be a major difference in the forcing.

Thank you for the suggestion. We have moved Fig. S5 to main figure (Fig. 6). Fig. R6 shows the fluctuations of observed SST and simulated SST in three sets of coupled and uncoupled model. There is no fluctuation as expected in uncoupled simulations, whereas the simulated SST fluctuates with phases similar to the observed at different locations. The amplitudes in ECHAM5-SIT and CAM5-SIT are similar to the observed, whereas those in HiRAM-SIT seems to be smaller in the western Pacific. The differences between models are likely due to the different atmospheric model configurations, because they were coupled to the same 1-D ocean model. Since the atmosphere is the main driver to extract heat form the ocean, different responses of atmospheric models likely have different effects on SST. The purpose of this study is to demonstrate the critical role of coupling in improving MJO simulations. The cause of quantitative differences between models needs further detailed analysis to pinpoint. We have added this discussion and Fig. R6 in the revised manuscript.

[Figure]

Fig R6. The SST (°C) with respect to MJO phases for intraseasonal anomalies (i.e., with 20–100-day filtering) in (a) observations and simulations by using the (b–d) coupled and (e–g) uncoupled AGCM. Observations are in suit with data from OISST.

24. Figure 6, please label the units.

It is revised.

25. Line 260-262, this sentence is for the discussion next section, right? If so, please change "We diagnosed" to "We next diagnosed".

It is revised.

26. Line 275, what do you mean by "smaller LH negative"?

Sorry for the typo. It is corrected to "negative LH bias".

27. Figure 7, can you comment on the residual term within the observation? Why is it larger than many other terms? Also, the budget analysis suggests the enhanced LH plays a major role on the correction. However, the only difference is the change of SST (coupled SST has a different value from the specified SST in the uncoupled simulation, is that correct?). How does the change of SST modulate the change of LH?

Large residual is a known issue even in the reanalysis. Kiranmayi and Maloney (2011) described the problem as follows: "A significant residual exists in the anomalous MSE budget, such that both reanalysis products appear to be missing or misrepresenting some MSE recharge process in advance of MJO convection. Such residuals have also been noted in other reanalysis products. For example, the NASA MERRA product misrepresents the moistening process in advance of MJO convection such that a large analysis increment must be added to the model humidity field to account for this missing source. Reanalysis data is derived by assimilation of observations with an analysis model. Thus, the reanalysis fields are dependent on model parameterizations and approximations and are not a perfect reconstruction of the real atmosphere." Our result indicates that it remains unsolved 10 years later. The discussion in Kiranmayi and Maloney (2011) about the large residual term in the observation seemingly remains valid. Further improvement in model, data assimilation, and observation are evidently needed.

Yes, SST in the coupled model has a different value from the specified SST in the uncoupled simulation. The SST in the coupled model is modulated by the responses of atmospheric model to the SST forcing. The atmospheric perturbations (e.g., surface wind, moisture and air temperature) near ocean surface in turn modify surface fluxes and SST. Over ocean surface, LH flux is usually the dominant term. LH flux represents the heat extracted from the ocean surface through evaporation and is basically determined by wind and air-sea moisture difference.

Kiranmayi, L., & Maloney, E. D. (2011). Intraseasonal moist static energy budget in reanalysis data. Journal of Geophysical Research: Atmospheres, 116(D21).

28. Figure 8, please label the units in the caption.

It is revised.

29. Line 288, Is this really the equatorial Kelvin wave? If so, can you clarify the wave speed of this Kelvin wave?

As discussed in the reply to comment 23, it does exhibit similar spatial characteristics to those of equatorial Kelvin wave. As demonstrated in previous studies, the Kelvin wave embedded in a MJO is coupled to the equatorial Rossby wave through deep convection. The coupled wave packet moves eastward in a speed comparable to the observed that is much slower than free Kelvin wave does.

30. Figure 9 and 10 suggest the dominant role of meridional advection moisture term. Does that imply the instantaneous SST horizontal distribution plays a key role on this change due to the coupling effect? Then, the change of varying moisture induces the intraseasonal circulation change.

Yes. Thank you for the nice interpretation. We have addressed more in the manuscript following your suggestion.

31. Figures S6 and S7 also suggest the background mean states are not the key contributor for the enhancement. Can we conclude that the change of SST distribution indeed the main driver for the enhancement? However, many other coupled models which cannot resolve the surface warm layer show the coupling with ocean may make the MJO simulation worse. Can you further quantify the key role of resolving the surface warm layer on the resulting SST which consistently change the models' boundary layer? Diurnal cycle of warm layer? Or others?

As observed in the MJO field campaign DYNAMO, ocean responded quickly to atmospheric forcing (Matthews et al., 2014). The findings indicate that the MJO is an atmosphere-ocean coupled phenomenon. It is therefore crucial for a

model being able to properly simulate the coupling process. Tseng et al. (2015) demonstrated using ECHAM5-SIT that increasing vertical resolution in the upper few meters resulted in quicker ocean temperature fluctuation in the MJO. It was proposed that an ocean model with high vertical resolution resolving warm layer responds almost immediately to atmospheric perturbations and therefore simulates more closely the coupled nature of the MJO. Jiang et al. (2015) in a MJO simulation comparison study revealed that models with coupling tended to outperform atmosphere-only models. It is not clear why certain model when coupled has poorer performance. It is likely case dependent. In our study, we clearly demonstrate that coupling improves the simulations in three AGCMs with very different configurations and parameterization schemes. It would be interesting to see whether similar improvement can be achieved in other models when similar high-resolution 1-D ocean model is implemented. We hope that our study could attract more studies working on this issue. We have added this discussion in the discussion.

32. Discussion section: the fundamental driver from the coupled air-sea interaction process is still unclear from this manuscript. However, that is the major point of bringing the coupling of resolving the ocean surface warm layer. The coupled results change the boundary layer through the SST update. Can the authors comment on this and provide some further guidelines how the modelers may improve the MJO simulation practically through this approach?

Tseng et al. (2015) suggests that a finer vertical resolution more effectively resolves temperature variations in the ocean warm layer and enhances atmospheric–ocean coupling, thus enabling the upper ocean to more efficiently respond to atmospheric forcing by providing sensible and latent heat fluxes. This results in superior synchronization between the lower atmosphere and the upper ocean. The coupling also enhances the moistening in the boundary layer and shallow convection that leads the deep convection, a key process identified for healthy eastward propagation of the MJO. Following this discussion, Lan et al. (2021) further suggests that better resolving the fine structure of the upper-ocean temperature and therefore the air–sea interaction leads to more realistic intraseasonal variability in both SST and atmospheric circulation.

The main indication here is that resolving of the warm layer in ocean model better simulates the intraseasonal signals. Our study suggests the effectiveness of air–sea coupling for improving MJO simulation in a climate model and demonstrated the importance of warm layer. The findings enhance our

understanding of the physical processes that shape the characteristics of the MJO.

We have added in the conclusion: "Our study suggested the effectiveness of air–sea coupling for improving MJO simulation in a climate model and demonstrated the importance of warm layer. The findings enhance our understanding of the physical processes that shape the characteristics of the MJO."

Hendon, H. H., Liebmann, B., and Glick, J. D.: Oceanic Kelvin waves and the Madden–Julian oscillation, Journal of the Atmospheric Sciences, 55, 88-101, 1998.

Lan, Y. Y., Hsu, H. H., Tseng, W. L., and Jiang, L. C.: Embedding a One-column Ocean Model (SIT 1.06) in the Community Atmosphere Model 5.3 (CAM5.3; CAM5–SIT v1.0) to Improve Madden–Julian Oscillation Simulation in Boreal Winter, Geosci. Model Dev. Discuss., 2021, 1-49, 10.5194/gmd-2021-346, 2021.

Matthews, A. J., Baranowski, D. B., Heywood, K. J., Flatau, P. J., and Schmidtko, S.: The surface diurnal warm layer in the Indian Ocean during CINDY/DYNAMO, Journal of Climate, 27, 9101-9122, 2014.

Tseng, W.-L., Tsuang, B.-J., Keenlyside, N., Hsu, H.-H., and Tu, C.-Y.: Resolving the upper-ocean warm layer improves the simulation of the Madden–Julian oscillation, Climate Dynamics, 1-17, 10.1007/s00382-014-2315-1, 2015.

Webber, B. G., Matthews, A. J., and Heywood, K. J.: A dynamical ocean feedback mechanism for the Madden–Julian oscillation, Quarterly Journal of the Royal Meteorological Society: A journal of the atmospheric sciences, applied meteorology and physical oceanography, 136, 740-754, 2010.